# Insight into the Gut–Brain Axis and the Productive Performance and Egg Quality Response to *Kudzu* Leaf Flavonoid Supplementation in Late-Laying Hens

**DOI:** 10.3390/ani14192780

**Published:** 2024-09-26

**Authors:** Shi Tang, Yaodong Hu, Jiahui Luo, Meijun Hu, Maolin Chen, Dehan Ye, Jingsong Ye, Fuguang Xue

**Affiliations:** 1College of Animal Science, Xichang University, Xichang 615000, China; tangshi1989103@163.com (S.T.); xcc20210231@xcc.edu.cn (Y.H.); 2College of Animal Science, Jiangxi Agricultural University, Nanchang 330045, China; 19354348202@163.com (J.L.); 18296186736@163.com (M.H.); cml121233@163.com (M.C.); 18070318412@163.com (D.Y.); 18479692205@163.com (J.Y.)

**Keywords:** antibiotic alternative, *kudzu* leaf flavonoids, yellow-feathered broilers, antioxidant, CECAL microbiota

## Abstract

**Simple Summary:**

Plant extracts contain a series of bio-active ingredients. Particularly, flavonoids exert efficient bio-activity and excellent microbial modulatory capacities, which are considered to be appropriate feed additives in husbandry production. In this study, flavonoids extracted from *kudzu* leaf (KL) was chosen to investigate their promotive effects on the productive performance and egg quality of layer hens. The results indicate that the *kudzu* leaf flavonoid (KLF) supplement significantly proliferated probiotics, such as *Bifidobacterium* sp. and *Lactobacillus* sp., which may have further interacted with hypothalamus genes, thus decreasing the deformity rate while increasing eggshell strength in the finishing phase. The findings indicate that KLF could be used as an effective feed additive for prolonging laying rates during the late-laying stage and may further improve feed efficiency, thus lowering costs.

**Abstract:**

(1) Background: Improving feed efficiency and the vitality of the reproductive system in the late stage of the egg-laying period is of great significance for prolonging the egg-laying cycle and improving egg quality. In the present study, a new flavonoid, which was extracted from *kudzu* leaf, was chosen to investigate its effects on the productive performance and egg quality of late-laying hens. (2) Methods: A total of 360 500-day-old Hy-Line Brown layer hens were randomly divided into a control treatment group (no KLF supplementation), and groups that received 0.2%, 0.4%, 0.6%, 0.8%, and 1.0% KLF supplement treatments. Each treatment contained 6 replicates, with 10 hens in each replicate. Productive performance metrics, including the daily egg production, egg weight, the number of deformed eggs, egg quality, egg density, egg shape index, eggshell strength, yolk color, and the Haugh unit, were meticulously recorded for each replicate. Furthermore, microbial communities and hypothalamus gene expressions were investigated based on the results of the productive performance and egg quality. (3) Results: KLF supplementation significantly decreased the deformity rate while significantly increasing the eggshell strength in the finishing phase afterward (*p* < 0.05). Specifically, hens supplemented with 0.6% KLF possessed the lowest deformed egg rate. KLF supplementation significantly increased the relative abundances of *Bifidobacterium* sp., *Blautia* sp., *Lactococcus* sp., and *Lactobacillus* sp., while significantly decreasing *Parasutterella* sp. and *Escherichia-Shigella* sp. (*p* < 0.05). Furthermore, the interactive analysis showed the hypothalamus gene expression mainly interacted with probiotics, such as *Bifidobacterium* sp. and *Lactobacillus* sp., through ribosome biogenesis, nucleocytoplasmic transport, and cAMP signaling pathways. (4) Conclusions: The findings of the present study indicate that KLF supplementation significantly proliferated probiotics, such as *Bifidobacterium* and *Lactobacillus*, which may have further interacted with hypothalamus genes, thus decreasing the deformity rate while increasing eggshell strength in the finishing phase.

## 1. Introduction

Improving the feed efficiency (FE) of layer hens to lower the feed-to-egg ratio and lower costs is currently the primary target for the sustainable production of layer hens [1]. However, the reproductive system of laying hens traditionally deteriorates after 60 weeks of age, which leads to a gradual decline in egg production performance and egg quality among large-scale cage-reared hens [2]. A limited range of activity in cage-reared hens decreases their intestinal degradability in the late-laying period, which may exacerbate feed consumption and causatively decline the laying rate. Therefore, exploring an effective method to improve the vitality of the reproductive system in the late-laying stage is of great significance for prolonging the egg-laying cycle and improving egg quality.

Bio-active ingredients extracted from Chinese herbal medicine, particularly flavonoids, efficiently regulate physiological processes with their high bio-activity and antioxidant capacities [3]. Research has shown that the proper amount of flavonoids supplemented to the feeding diet abundantly proliferate intestinal microbiota diversity and further promote nutrient digestibility and absorption, which effectively improve egg qualities [4,5,6]. Similar results could also be found in our previous study; flavonoids extracted from *kudzu* leaf (KLF) significantly improved the relative abundance of microbiota diversity and probiotic content in broilers, which resulted in the enhancement of feed efficiency [7]. However, whether KLF supplementation could work for prolonging laying performance and its underlying mechanism are still unclear.

Compared to single-focus studies on gastrointestinal microbiota communities, the interactive effects between the host and microbiota provided deeper insights into the underling mechanism in regulating physiological metabolism. The mutually beneficial correlation between the host and the gut microbiota was the focus of recent studies that examined the host’s provision of hospitable niches and undigested food for the microbiome and the gut microorganisms’ feedback to the host through secondary metabolites and neuroactive components [8,9]. Further investigations have pointed out the significant bidirectional gut–brain interactions, such as the modulated physiological and gastrointestinal homeostatic functions through the information signaling pathways between the central nervous system and gut-based enteroendocrine cells [10,11,12]. However, information about whether KLF supplementation impacts the interactions of the gut–brain axis and the underlying pathways is still lacking. 

Therefore, in the present study, flavonoid extracted from *kudzu* leaf was applied to investigate its underlying mechanism and effects on regulating the laying rate and egg quality. We hypothesized that supplementing with KLF would modulate the bacterial communities and proliferate probiotic diversity, which would further enhance microbe-derived signals to alterations in hypothalamic gene expression and digestion-related enzymatic catabolism and eventually improve the productive performance in late-laying hens.

## 2. Materials and Methods

The animals and trial procedures used in the present study were in accordance with the recommendations of the academy’s guidelines for animal research and approved by the Animal Ethics Committee of Xichang University; the approval code is XC20240309.

### 2.1. Kudzu Leaf Flavonoid Preparation

*Kudzu* leaf flavonoids (KLFs) were extracted using an ultrasonic (Elmasonic X-tra Flexl; Elma, Konstanz, Germany) extracting procedure. In brief, the solid-to-liquid ratio between the mushed *kudzu* leaf and 57.0% ethanol was set to 1:30.5, which underwent a 29.88 min-long extraction process. The extractions were further centrifugated at 3000 rpm for 10 min. The post-extracted liquor was filtered after cooling and prepared for supplementation in the feed of layer hens.

The flavonoid content was further measured based on the GB/T 42114-2022 [13]. In brief, 10 mg of a flavonoid standard sample (*n* = 6) was dissolved in 50 mL of ethanol, followed by a gradual collection of 1, 2, 3, 4, 5, and 6 mL of the standard solution in a 50 mL volumetric flask, and a 15 mL addition of 95% ethanol. Furthermore, 1 mL of 100 g/L aluminum nitrate solution and 1 mL of 9.8 g/L potassium acetate solution were added to the mixture. The absorption of each standard solution was measured at 415 nm wavelength after standing for 60 min. The standard curve was acquired as follows:y = 0.4145x + 0.001(R^2^ = 0.9997)
where y = flavonoids content, and x = absorption.

### 2.2. Experimental Design

A total of 360 500-day-old Hy-Line Brown layer hens were randomly divided into a control treatment group (no KLF supplementation), and groups that received 0.2%, 0.4%, 0.6%, 0.8%, and 1.0% KLF supplementation treatments. Each treatment contained 6 replicates, with 10 hens in each replicate. The study lasted for 135 days, which consisted of a 15-day-long pre-feeding period and a 120-day-long main feeding period. The main feeding period was divided into two stages, each containing 60 days.

The composition and nutritional levels of the experimental diets are detailed in Table 1. The research was conducted on chickens from Xichang University, and the laying hens were housed in a three-tier cage system. Each hen was reared in an individual cage, with free access to water and a daily feed allowance of 110 g. A combination of natural and artificial lighting was provided to achieve 16 h of daily light exposure, and the indoor temperature ranged from 20 to 26 °C.

### 2.3. Productive Performance and Egg Quality Measurements

Productive performance metrics, including the daily egg production, total egg weight, and the number of deformed eggs, were meticulously recorded for each replicate during the feeding period. In addition, the remaining feed mass was measured to determine the egg production rate (EPR), deformity rate (DR), average egg weight (AEW), average daily feed intake (ADFI), and feed-to-egg ratio (FER) to determine the benefits of KLF supplementation on feed efficiency. The metrics were acquired with the following equations:EPR = Daily egg number/total chicken number × 100% 
DR = Deformed egg number/Daily egg number × 100%
AEW = total daily egg weight/Daily egg number × 100%
FER = ADFI/AEW

Subsequently, egg quality was assessed on the 60th and 120th days. For the egg quality measurements, 10 eggs from each replicate, for a total of 360 eggs, were collected. Parameters such as the egg weight, egg specific gravity, egg shape index, eggshell strength, Haugh unit, and yolk ratio were evaluated. The results of each parameter were recorded as the average of each replicate and presented as the mean ± SEM.

An egg quality analyzer (05-UM-01, Nanjing Ouxi Science and Trade Co., Ltd, Nanjing, China) was utilized and the saltwater flotation method was employed to determine the egg specific gravity. A vernier caliper (SF2000, Guilin Guanglu Digital Measurement and Control Co., Ltd, Guilin, China) was used to measure the long and short diameters of the eggs for calculating the egg shape index.

### 2.4. Hypothalamus Acquisition and Transcriptomic Measurement

The optimum KLF supplement treatment was chosen for further transcriptomic and metagenomic analysis based on the results of productive performance and egg quality parameters. The hypothalamus from one bird per replication was sampled within 20 min of slaughter and dissected from the other brain parts using mini tweezers, according to the markers described by Xu et al. [14]. The separated hypothalamic tissues were immediately frozen in liquid nitrogen before storage at −80 °C for further transcriptome analysis. RNA was isolated from each sample using an RNA kit (Takara, Dalian, China), according to the manufacturer’s recommendations, and purified by an Agencourt^®^ RNAClean™ XP (Beckman Coulter, Inc., Indianapolis, IN, USA). Subsequently, the FastQC program was used for quality control, and reads with quality scores lower than 20 were removed. Alignment and estimation of the gene expression levels was performed using the transcriptome as a reference, and differentially expressed gene statistics were obtained using DESeq2 (version 1.42.0) software. The Cluster Profiler package (version 3.14.3) was used for GSEA analysis to detect expression changes, and genes with an adjusted *p* < 0.05 (*p*. adj < 0.05) and a fold change (FC) ≥ 2.0 were included for further Gene Ontology (GO) and Kyoto Encyclopedia of Genes and Genomes (KEGG) pathway enrichment analyses.

### 2.5. Gastrointestinal Microbiota Determination

Cecal samples of the CON and the optimal KLF supplementation treatments were collected from one bird per replication on the last day of the trial. The DNA from each sample was extracted according to the CTAB/SDS method [15], followed by amplification of the 16S rRNA gene V4 region using primer pairs 515F and 806R (F: GTGCCAGCMGCCGCGGTAA and R: GGACTACVSGGGTATCTAAT) [16]. Samples with a bright main strip between 400 and 450 bp were chosen to generate sequencing libraries using the TruSeq^®^ DNA PCR-Free Sample Preparation Kit (Illumina Inc., San Diego, CA, USA). Their library quality was further assessed on the Qubit@ 2.0 Fluorometer (Thermo Scientific (China) Co., Ltd., Shanghai, China) and Agilent Bioanalyzer 2100 system (Agilent Technologies, Inc., Palo Alto, CA, USA). The Illumina HiSeq 4000 platform (Illumina Inc., San Diego, CA, USA) was used to obtain the sequencing data. Sequences identified to be >97% similar were assigned into the same operating taxonomic units (OTUs), which received further taxonomic annotation using the Green Gene Database (https://github.com/biocore/greengenes2, accessed on 16 April 2024). Subsequent analyses of the alpha diversity, beta diversity, and functional prediction were performed based on the OTU results.

### 2.6. Statistical Analysis

Differential analyses of the productive performance and egg quality parameters were verified through a normal distribution test using the SAS (SAS Institute, Inc., Cary, NC, USA) procedure “proc univariate data = test normal”. Subsequently, a comparison test was carried out using one-way ANOVA, and the Student–Newman–Keuls model was subsequently used for the post hoc test to investigate the differences among the treatments. All results are presented as the mean ± SEM.

Microbial alpha diversity analysis, which included the Chao1, Shannon, Simpson, and ACE indexes were applied to analyze the complexity of species diversity, while the beta diversity, functionally evaluating differences in the species complexity among treatments, was calculated with QIIME (Version 1.7.0) and displayed using R software (Version 3.15.3, R Core Team, Vienna, Austria). Principle coordinate analysis (PCoA) was constructed using the WGCNA package, stat packages, and the ggplot2 package in R software. Spearman correlations among the production performance, immune organ indexes, and bacteria communities were assessed using the PROC CORR procedure in SAS 9.2, and then a correlation matrix was created and visualized in a heatmap format using R software. For all differential analysis results, a *p*-value < 0.05 was considered to be significant, and 0.05 ≤ *p* < 0.10 was considered to be a tendency.

## 3. Results

### 3.1. Effects of Kudzu Leaf Flavonoid Supplements on Productive Performance and Egg Quality

Productive performance metrics, including the average daily feed intake (ADFI), laying rate, feed/egg ratio, and deformity rate, were primarily measured, and the results are shown in Table 2. KLF supplementation showed no effects on the ADFI, feed/egg ratio, laying rate, and deformity egg rate in the first phase (*p* > 0.05). However, the deformity rate showed a significant decrease in the finishing phase after KLF supplementation (*p* < 0.05), and supplementing with 0.6% KLF showed the lowest deformity egg rate. No other significant alterations were detected in the ADFI, feed/egg ratio, and laying rate after KLF supplementation in the finishing phase (*p* > 0.05).

Egg quality was measured for the purpose of investigating the effects of KLF supplement, and the results are shown in Table 3. Egg quality results show no significant alterations in the first feed phase after KLF supplementation (*p* > 0.05). However, the eggshell strength showed a significant increase after 0.6% KLF supplementation (*p* < 0.05), and the Haugh unit showed an increasing trend with increasing KLF supplementation (0.05 < *p* < 0.10). No other significant alterations were detected in egg weight, relative egg density, or egg shell thickness. Based on the results of the productive performance and egg quality parameters, layer hens that underwent the 0.6% KLF supplement treatment were selected for subsequent gene and microbial analysis.

### 3.2. Effects of KLF Supplementation on Hypothalamus Gene Expression of Late-Laying Hens

The transcriptomic sequencing results of the hypothalamus gene expressions are all displayed in Appendix A. Altogether, a total of 9834 mRNAs were identified between the KLF- and CON-treated birds. All genes were comprehensively differentially expressed using a quality filtering process, and the results are shown in Figure 1.

Principal component analysis (PCA) was carried out on all filtered genes to determine the comprehensively differentially expressed genes. As shown in Figure 1A, PC1 and PC2 accounted for 46.8% and 18.7% of the total variation, respectively. The gene expressions showed significant differences between the KLF- and CON-treated birds according to PC1 and PC2.

Subsequently, differential analysis on the hypothalamus gene expression was conducted, and the DESeq2 method was used to identify the differentially expressed (DE) mRNAs in the hypothalamus between the KLF- and CON-treated layer hens. A volcano plot analysis is shown in Figure 1B. Based on the results, a total of 124 DE mRNAs were identified, which included the 67 up-regulated and 57 down-regulated genes in the KLF treatment hypothalamus compared to CON based on a threshold FDR of ≤0.05 and |log2FC| ≥ 1. All these genes are shown in Appendix A and were selected for subsequent functional analysis.

Functional analyses, including GO and KEGG pathway analyses, were applied, and the results are shown in Figure 2. The GO results indicate that the DE genes were mainly enriched in cellular processes, biological regulation, metabolic processes, binding, catalytic activity, and cell parts (Figure 2A). Specifically, the upregulated genes in the KLF treatment mainly enriched in cellular processes and biological regulation. In addition, the results of the pathway analysis of the differentially expressed genes are shown in Figure 2B. Ribosome biogenesis, nucleocytoplasmic transport, and cAMP signaling pathways were significantly enriched based on the significantly differentially expressed genes between the KLF and CON treatments.

### 3.3. Effects of KLF Supplementation on Gastrointestinal Microbiome

Modulatory effects of the KLF supplement treatment on the gastrointestinal microbiome were subsequently investigated, and the results are shown as follows.

In total, the microbiota sequencing identified 4800 OTUs, 12 phyla, and more than 290 genera after quality control. All taxonomic information is displayed in Appendix A. All identified bacteria were chosen for α-diversity parameter analysis, and the results are shown in Table 4. Based on the results, a significant increase was found in the Chao1 index in the KLF-treated hens compared to the CON hens (*p* < 0.05). No other significant alterations were observed between the KLF and CON treatments.

PCoA analysis, which clarified the monolithic discrepancy in the microbial profiles between the KLF and CON treatments, was performed to assess the β–diversity. As shown in Figure 3, PCoA axes 1 and 2 accounted for 46.91% and 36.38%, respectively. The bacterial communities in the KLF treatment were significantly different from those in the CON group, indicating a significant alteration in the composition of the gut microbiota under the KLF supplement treatment.

Differential analysis of the relative abundances of gut bacteria was performed to investigate the effects of KLF supplementation on gastrointestinal micro-ecosystem; the results are shown in Table 5. At the genera level, *Bacteroides*, *Alistipes* sp., *Megamonas* sp., *Barnesiella* sp., and *Faecalibacterium* sp. were the five most abundant genera in all the treatments. KLF supplementation significantly increased the relative abundances of *Bifidobacterium* sp., *Blautia* sp., *Lactococcus* sp., *and Lactobacillus* sp. but significantly decreased *Parasutterella* sp. and *Escherichia-Shigella* sp. (*p* < 0.05). Specifically, probiotics such as *Bifidobacterium* sp., and *Lactobacillus* sp. showed significant proliferation after KLF supplementation compared to the CON group (*p* < 0.05). No other significant alterations were found in the relative abundances of gut microbiota between the KLF and CON treatments.

Functions presented in the differentially identified microbiota were predicted using Tax4Fun (http://tax4fun.gobics.de/, accessed on 16 April 2024), and the results are shown in Figure 4. Metabolic processes, mainly including carbohydrate metabolism, amino acid metabolism, and cofactors, were the predominant functional pathways. Specifically, the functions of proliferated bacteria were mostly enriched in carbohydrate and amino acid metabolism and may further provide higher energy for the laying process.

### 3.4. Interaction Effects among Hypothalamus Genes, Hormones, and Cecal Microbiota

The interaction effects between gut bacteria and hypothalamus genes were analyzed to determine the underlying gut–brain correlations regulating the productive performance of layer hens. The results are shown in Figure 5.

The hypothalamus genes showed low correlations with the bacterial communities, except for the significant correlation between hypothalamus genes and *Lactobacillus* sp. and *Parasutterella* sp.

Specifically, *Lactobacillus* sp. was significantly correlated with *IRS1*, while *Parasutterella* sp. was significantly correlated with *PRKAG2*. In addition, *IRS1* showed positive correlations with *PRKAG2*, *PPP2R3C*, *PPM1K*, and *GLUL. PPP2R3C* was significantly positive correlated with *GADD45B*, while negatively correlated with *ZDHHC2* and *GLUL*. No other significant correlations were detected among the genes.

## 4. Discussion

Productive performance metrics significantly impact the sustainable production of layer hens and are modulated by a certain number of factors, including feed intake, digestibility, nutrient absorption, and physiological status, which are uniformly regulated by the brain instructions [17,18,19]. Plant-extracted flavonoids exert effective free radical scavenging capabilities, coupled with easy acquisition properties, making them a valuable asset in husbandry production. The findings of the present study indicate that the supplementation of KLF significantly decreased the deformity rate and boosted the egg shell strength of late-laying hens. The reasons are as follows.

### 4.1. Modulatory Effects on Gut Microbial Communities in Promoting Productive Performance

In traditional practices, increased abundances of gut microbial communities and the proliferated probiotic abundances significantly promote gastrointestinal digestibility and the nutrient degrading ability [20,21], which may further improve the laying rate of hens. The microbiota in the cecum exhibited high metabolic activity in the gastrointestinal tract of chickens, and higher cecal bacteria diversity provided more efficient intestinal digestibility [22] and promoted feed utilization. In this study, KLF supplementation significantly increased the bacterial α-diversity and probiotic communities and functionally induced the carbohydrate-degrading process, which may further provide more abundant substrates for nutrient synthesis and higher energy for the laying process [23]. Additionally, the relative number of probiotics significantly increased after KLF supplementation, positively interacting with the intestinal epithelium and enhancing intestinal digestibility [24,25]. The increased probiotics further provide evidential support for the enhanced productive performance of chickens after KLF supplementation.

Moreover, flavonoid supplementation may also improve egg quality by promoting body health, benefitting the intestinal health of layer hens [26] Flavonoids are generally known to benefit animal health through the enhancement of body immunity and anti-oxidant capacity. These beneficial capacities may be the cause of the increase in egg quality. As a previous study showed, the addition of flavonoids increased the trans-epithelial electrical resistance and stimulated the immune system response by enhancing the phagocytic activity of monocytes [27]; thus, the reproductive system was improved and the eggshell strength was enhanced.

### 4.2. Bidirectional Gut–Brain Interactions to Boost Productive Performance

Bidirectional gut–brain interactions have attracted considerable attention in recent microbial studies, in addition to the significant regulatory effects on physiological and gastrointestinal homeostatic functions [10,28,29]. In this study, it was found that interactions also occurred during the KLF treatment process and may be the underlying mechanism in regulating productive performance.

KLF significantly modulated intestinal microbial communities, which further altered the nutritional exchange process through the epithelium. Changes in the intestinal permeability create a passage for microbiota and their metabolites, which are transported from the lumen to the central nervous system, directly via the systemic circulation or indirectly by interacting with receptors on gut-based enteroendocrine cells [30], which further guide the ingestive behavior and digestibility.

Several aspects, such as higher leptin and insulin concentrations, account for feeding behavior and feed efficiency [28]. Additionally, the majority (>95%) of 5-hydroxytryptamine(5-HT) was synthesized via sequential conversion of tryptophan by the commensal bacteria belonging to *Blautia* and *Lactobacillus*. The significant proliferation of *Blautia* and *Lactobacillus* in response to KLF supplementation may ultimately increase the 5-HT content and improve brain excitability and physiological nutrient absorption. In addition, according to the results shown in Figure 5, *Lactobacillus* sp. was significantly correlated with *IRS1*. This correlation may further promote productive performance, as nutritional exchanges are complicated processes that require the assistance of energy and ion channels. The up-regulated genes of *ATP7A*, *ATP2A1*, protein kinase, AMP-activated gamma 2 (*PRKAG2*), and insulin receptor substrate 1(*IRS1*) after KLF supplementation may provide more energy for nutrient-exchanging processes, thus enhancing the productive performance metrics.

Furthermore, the hypothalamus is the central instruction-issuing organization, and the altered expressions of hypothalamus genes detected in the present study may further alter the physiological conditions to adapt to the supplementation of KLF, subsequently inducing changes in the productive performance metrics.

Certain characteristics are considered to be positively correlated with health physiological conditions and better productive performance metrics, such as a high level of microbial diversity, a favorable amount of butyrate-producing bacteria, and a better ability to withstand pathogens [31,32]. In the present study, probiotics, including *Bifidobacterium* sp., *Lactococcus* sp., and *Lactobacillus* sp., significantly (*p* < 0.05) proliferated after KLF supplementation. Probiotics, such as the *Bifidobacterium* sp. and *Lactobacillus* sp., whose products include IgA, contributed to the prevention of pathogen colonization [33]. These changes subsequently restrict the relative abundances of pathogens and may further strengthen epithelial barrier functions, which may subsequently promote gastric degradability and improve feed efficiency.

Although the promotive effects of KLF were significantly demonstrated in late-laying hens, further investigations are required to determine the modulatory mechanism of KLF. Functional analysis of the significantly proliferated bacterial communities should be carried out using the cultivation method, and functions of the differentially expressed genes should be studied using the cell culture method. These additional studies would help verify the modulatory effects of KLF.

## 5. Conclusions

The findings of the present study indicate that KLF supplementation effectively improved the productive performance and egg quality through significantly proliferated probiotics, such as *Bifidobacterium* sp. and *Lactobacillus* sp., which may further interact with hypothalamus genes, thus promoting body health and productive performance.

## Figures and Tables

**Figure 1 animals-14-02780-f001:**
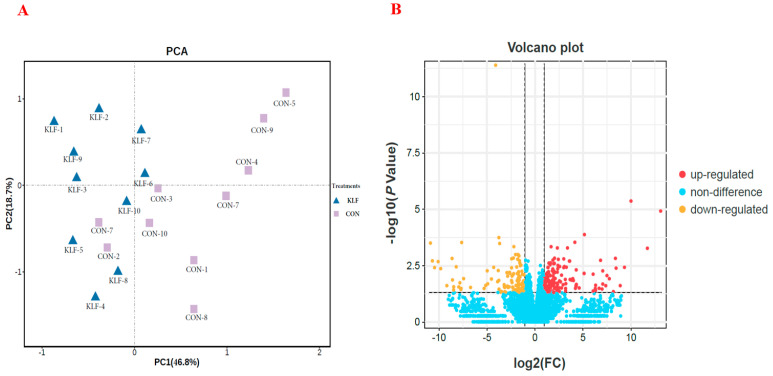
Differential analysis of the relative gene expressions of the hypothalamus between the KLF supplement treatment and the control treatment. CON = control treatment; KLF = *kudzu* leaf flavonoid supplement treatment. (**A**) Principal component analysis (PCA) of the relative gene expressions of the hypothalamus between the KLF supplement treatment and the control treatment. (**B**) Volcano plot of the differentially expressed genes between the KLF supplement treatment and the control treatment.

**Figure 2 animals-14-02780-f002:**
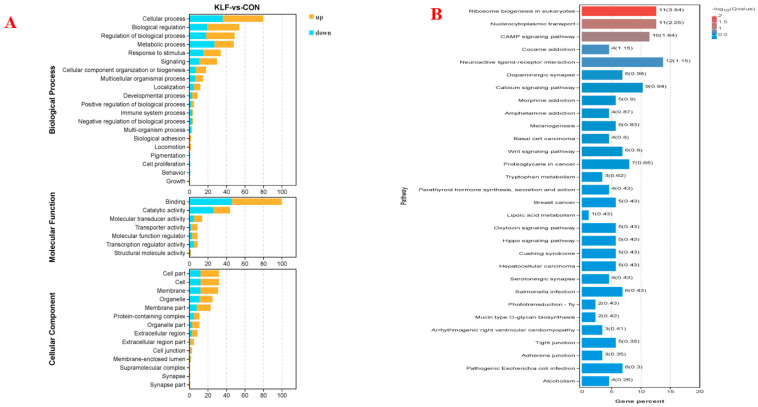
Pathway enrichment analysis of the differentially expressed genes between the KLF supplement treatment and the control treatment. (**A**) Gene ontology (GO) enrichment analysis of differentially expressed genes between the KLF supplement treatment and the control treatment. (**B**) KEGG pathway enrichment analysis of differentially expressed genes between the KLF supplement treatment and the control treatment.

**Figure 3 animals-14-02780-f003:**
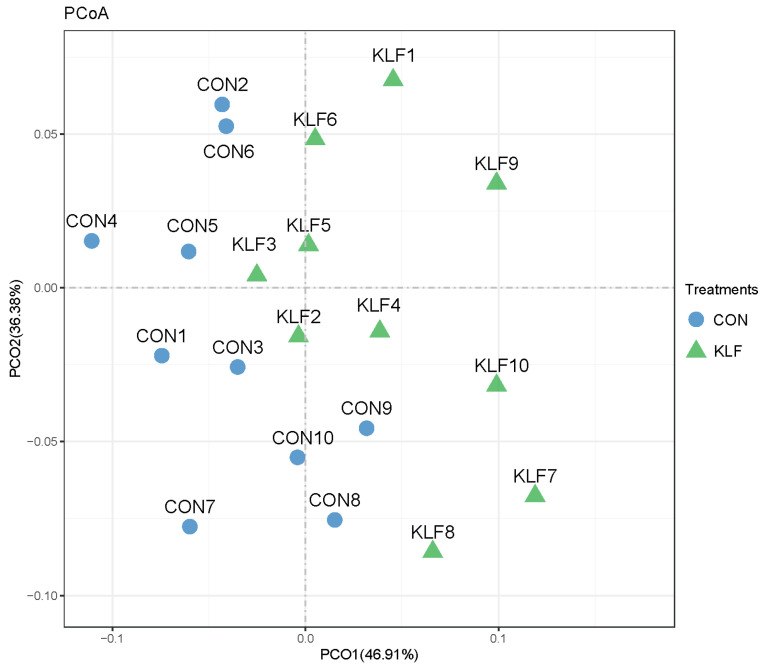
Principal coordinate analysis (PCoA) of community structures of the gut microbial community between the KLF and control treatments. CON = control treatment; KLF = *kudzu* leaf flavonoid treatment.

**Figure 4 animals-14-02780-f004:**
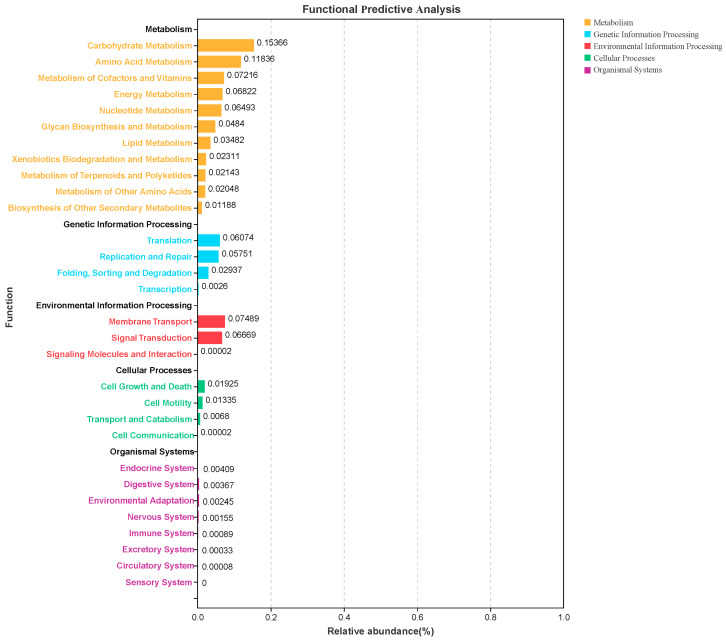
Functional prediction analysis of the significantly altered bacterial communities between the KLF and CON treatments.

**Figure 5 animals-14-02780-f005:**
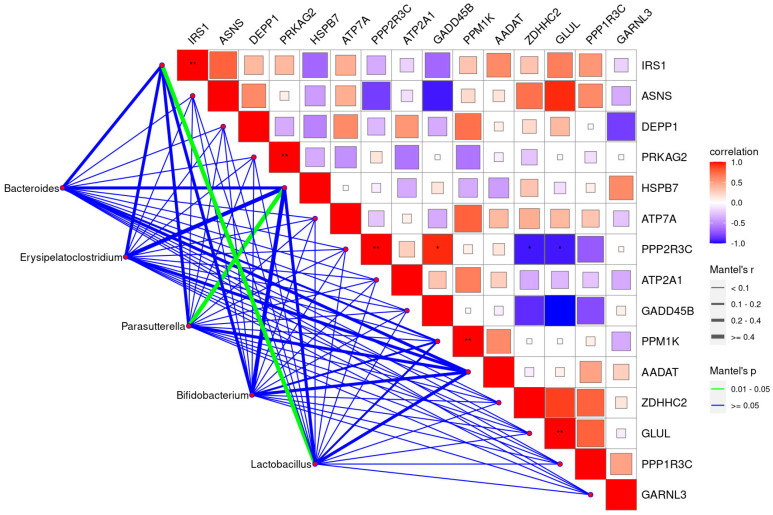
Correlation analysis between relative abundances of gut bacteria and gene expressions in the hypothalamus. The red color represents a positive correlation while the blue color represents a negative correlation. “*” means a significant correlation (|r| > 0.55, *p* < 0.05), “**” means a significant correlation (|r| > 0.75, *p* < 0.01). Blue lines between the gut microbial communities and genes represent no significant correlation; green lines indicate significant correlations between gut microbial communities and hypothalamus genes.

**Table 1 animals-14-02780-t001:** Ingredient and chemical composition of experimental diets.

Ingredient	Composition Amount (%)
Corn	60.10
Soybean meal (SBM, CP 43%)	25.00
Soy oil	1.30
CaCO_3_	9.00
Calcium hydrophosphate (2 water) DCP	1.00
Salt	0.40
Levogyration- Lys-HCL (L- Lys-HCL, 98%)	0.10
Dextrorotation and levogyration-Met (DL-Met)	0.10
Primix ^a^	3.00
Total	100
Metabolizable energy (ME/MJ × kg^−1^)	11.31
Crude protein (CP)	15.30
Crude fat (EE)	3.07
Crude fiber (CF)	2.68
Calcium (Ca)	3.45
Phosphorus (P)	0.43
dLys	0.80
dMet	0.35
dCys	0.28
dM + C	0.63

^a^ Premix content: VA (12,000 IU/kg); VD (3950 IU/kg); VE (18 IU/kg); VK (8 mg/kg); VB1 (0.6 mg/kg); VB2 (4.8 mg/kg); VB6 (1.8 mg/kg); VB12 (10 mg/kg); folic acid (0.15 mg/kg); niacinamide (30 mg/kg); pantothenic acid (10.5 mg/kg); choline (480 mg), Fe (80 mg), Cu (8 mg), Mn (80 mg), Zn (60 mg), Se (0.15 mg), I (0.35 mg).

**Table 2 animals-14-02780-t002:** Effects of *kudzu* leaf flavonoid supplements on productive performance metrics of late-laying hens.

	Items	CON	0.2% KLF	0.4% KLF	0.6% KLF	0.8% KLF	1.0% KLF	SEM	*p*-Value
Initial Phase (d1–d60)	ADFI (g)	109.14	109.36	109.03	109.09	109.14	109.05	0.74	0.993
	Feed/egg ratio	1.89	1.87	1.85	1.82	1.82	1.84	0.11	0.285
	Laying rate (%)	82.4	82.57	83.46	83.45	82.79	82.29	0.93	0.234
	Deformity egg rate (%)	3.59	3.25	3.11	3.15	3.23	3.22	0.24	0.082
Finishing Phase (d61–d120)	ADFI (g)	109.43	109.46	109.09	109.03	109.07	109.16	0.84	0.976
	Feed/egg ratio	2.01	1.95	1.96	1.93	1.98	1.98	0.13	0.165
	Laying rate (%)	76.4	77.57	77.96	78.34	77.88	77.69	0.86	0.104
	Deformity egg rate (%)	5.69 ^a^	4.85 ^b^	4.77 ^b^	4.65 ^b^	4.70 ^b^	4.72 ^b^	0.21	0.042

Letters a and b in each row describe significant differences among the treatments at *p* < 0.05. CON = control treatment; KLF = *kudzu* leaf flavonoid supplement treatment; ADFI = average daily feed intake; SEM = standard error of mean.

**Table 3 animals-14-02780-t003:** Effects of *kudzu* leaf flavonoid supplements on egg quality of late-laying hens.

	Items	CON	0.2% KLF	0.4% KLF	0.6% KLF	0.8% KLF	1.0% KLF	SEM	*p-*Value
	Egg weight (g)	64.35	64.06	64.36	64.89	64.76	64.56	1.68	0.591
Initial Phase	Relative egg density	1.10	1.12	1.13	1.13	1.12	1.10	0.04	0.628
	Egg shell thickness (mm)	0.39	0.39	0.41	0.39	0.41	0.39	0.02	0.375
	Egg shape index	1.31	1.32	1.32	1.33	1.31	1.31	0.03	0.809
	Eggshell strength (N)	35.74	36.49	36.24	36.70	36.44	36.78	1.34	0.213
	Haugh unit	80.14	80.88	80.53	80.65	80.33	80.50	3.26	0.674
	Yolk percentage (%)	26.94	27.89	27.57	27.71	27.79	27.57	0.31	0.462
Finishing Phase	Egg weight (g)	65.04	65.27	65.13	65.00	65.28	65.19	2.31	0.671
	Relative egg density	1.10	1.10	1.11	1.12	1.12	1.11	0.04	0.568
	Egg shell thickness (mm)	0.39	0.40	0.41	0.41	0.39	0.40	0.02	0.465
	Egg shape index	1.31	1.33	1.31	1.33	1.32	1.31	0.03	0.749
	Eggshell strength (N)	33.94 ^b^	34.05 ^b^	34.74 ^ab^	35.36 ^a^	34.29 ^b^	34.63 ^ab^	1.06	0.034
	Haugh unit	74.14	75.66	76.06	75.96	75.47	75.45	0.76	0.095
	Yolk percentage (%)	27.24	27.36	27.60	27.42	27.69	27.62	0.36	0.346

The letters a and b in each row describe significant differences among the treatments at *p* < 0.05. CON = control treatment; KLF = *kudzu* leaf flavonoid supplement treatment; SEM = standard error of mean.

**Table 4 animals-14-02780-t004:** Effects of *kudzu* leaf flavonoid supplements on gut bacteria α-diversity parameters of late-laying hens.

Items	CON (*n* = 10)	KLF (*n* = 10)	SE	*p*-Value
Shannon	8.85	8.96	0.10	0.182
Simpson	0.98	0.98	0.00	0.342
Ace	1056.5	1071.4	52.0	0.418
Chao1	1156.3 ^b^	1216.6 ^a^	31.2	0.042
observed_species	934.9	972.5	26.8	0.317

The letters a and b in each row describe significant differences between the treatments at *p* < 0.05. CON = control treatment; KLF = *kudzu* leaf flavonoid supplement treatment; SE = standard error.

**Table 5 animals-14-02780-t005:** Effects of *kudzu* leaf flavonoid supplements on the relative abundances of gut microbiota at the level of genera.

Items	CON	KLF	SE	*p*-Value
*Bacteroides*	23.21 ^b^	28.10 ^a^	1.65	0.008
*Alistipes*	15.82	14.06	1.43	0.563
*Megamonas*	9.52	8.86	1.51	0.435
*Barnesiella*	6.39	6.60	1.11	0.705
*Faecalibacterium*	6.07	5.66	0.89	0.562
*Phascolarctobacterium*	5.73	6.94	0.57	0.102
*Eubacterium*	4.11	3.92	0.44	0.887
*Ruminococcus*	3.01	2.30	0.38	0.443
*Erysipelatoclostridium*	2.99 ^a^	1.18 ^b^	0.35	0.002
*Ruminococcaceae*	3.57	3.62	0.75	0.528
*Parasutterella*	2.33 ^a^	1.44 ^b^	0.16	0.002
*Desulfovibrio*	2.39	2.52	0.87	0.144
*Fournierella*	1.20 ^b^	1.42 ^a^	0.12	0.005
*Subdoligranulum*	0.92	0.62	0.16	0.540
*Bifidobacterium*	0.41 ^b^	1.31 ^a^	0.07	0.005
*Parabacteroides*	0.56	0.71	0.19	0.362
*Ruminiclostridium*	0.94	0.60	0.40	0.184
*Sellimonas*	0.76	0.37	0.06	0.078
*Oscillospira*	0.70	0.39	0.07	0.223
*Blautia*	0.84 ^a^	0.32 ^b^	0.08	0.018
*Butyricicoccus*	0.41	0.38	0.12	0.973
*Oscillibacter*	0.52	0.33	0.19	0.252
*Escherichia-Shigella*	0.49 ^a^	0.17 ^b^	0.05	0.025
*Lactococcus*	0.03 ^b^	0.07 ^a^	0.004	0.011
*Lactobacillus*	0.01 ^b^	0.04 ^a^	0.001	0.003
*Streptococcus*	0.01	0.01	0.004	0.795
others	7.10	8.08	0.42	0.073

The letters a and b in each row describe significant differences between the treatments at *p* < 0.05. SE = standard error; CON = control treatment; KLF = *kudzu* leaf flavonoid supplement treatment.

## Data Availability

The data presented in the study are deposited in the NCBI Sequence Read Archive (SRA, http://www.ncbi.nlm.nih.gov/Traces/sra/), accessed on 9 August 2021, accession number PRJNA753017.

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
