# Peer review of "Insight into the Gut–Brain Axis and the Productive Performance and Egg Quality Response to Kudzu Leaf Flavonoid Supplementation in Late-Laying Hens"

_animals, 2024, doi:10.3390/ani14192780_

Round 1

Reviewer 1 Report

Comments and Suggestions for Authors

Dear Authors, your paper is well written and the design is appropriate. I would like to ask you to include the statistical model(s) used and the comparison test (Tukey-Kramer, Dunnett?) Please include it in the statistical description in the materials and methods section, and include it also in the footnote of the pertinent tables. 

Best regards,

Author Response

Dear Authors, your paper is well written and the design is appropriate. I would like to ask you to include the statistical model(s) used and the comparison test (Tukey-Kramer, Dunnett?) Please include it in the statistical description in the materials and methods section, and include it also in the footnote of the pertinent tables. 

Best regards,

Re: Thank you for the comment and suggestions. Productive performances including egg production rate (EPR), deformity rate (DR), average egg weight (AEW), average daily feed intake (ADFI), and feed-to-egg ratio (FER)with the following equations:

EPR = Daily egg number / total chicken number×100%

DR = Deformed egg number / Daily egg number ×100%

AEW = total daily egg weight / Daily egg number ×100%

FER= ADFI / AEW

The results of egg quality parameters recorded by the average of each replicate, and calculated as Mean ± SEM

“the comparison test was applied using one-way ANOVA, and the Stu-dent-Newman-Keuls model was following used for the post-hoc test to investigate the differences among treatments ”

Please check Line 126-130. Line 183-185.

Reviewer 2 Report

Comments and Suggestions for Authors

Dear authors!

The article submitted for review is devoted to an important topic « Gut-brain axial insight into the productive performances and egg quality responses to kudzu‑leaf flavonoids supplement of late-laying hens».

The article has all the necessary sections and corresponds to the profile, goals and objectives of the journal.

I would like to note only a small remark on the data analysis in the article.

At the part «Materials and Methods» there is no clear data on which groups of birds were investigated for the parameters of microbial and hypothalamus gene expression. Only in the section introduction is information provided that microbial and hypothalamus gene expression were investigated base on the results of productive performance and egg quality.

At the discussion part there is no information on the section with results 3.4. Interactive effects among hypothalamus genes, hormones, and cecal microbiota. I recommend describe information at these results about hypothalamus genes correlate with the number of bacteria in the bird's intestine. Specifically, what are the genes IRS1, PRKAG2, PPP2R3C, PPM1K, GLUL, GADD45B, ZDHHC2. Have such correlations been reported before?

I recommend the article for publication after revision in these parts.

Author Response

  1. The article submitted for review is devoted to an important topic « Gut-brain axial insight into the productive performances and egg quality responses to kudzu‑leaf flavonoids supplement of late-laying hens».

The article has all the necessary sections and corresponds to the profile, goals and objectives of the journal.

I would like to note only a small remark on the data analysis in the article.

At the part «Materials and Methods» there is no clear data on which groups of birds were investigated for the parameters of microbial and hypothalamus gene expression. Only in the section introduction is information provided that microbial and hypothalamus gene expression were investigated base on the results of productive performance and egg quality.

Re: Thank you for the comment and sorry for not providing the accuracy description. We added” The optimum KLF supplement treatment was chosen for further transcriptomic and metagenomic analysis based on the results of productive and egg quality parameters. ” in line 145-146. And in the results section, sentence of “Based on the results of productive and egg quality parameters, layer hens in 0.6% KLF supplement treatment were selected for the following gene and microbial analysis. ” was added. Please check line 217-219.

At the discussion part there is no information on the section with results 3.4. Interactive effects among hypothalamus genes, hormones, and cecal microbiota. I recommend describe information at these results about hypothalamus genes correlate with the number of bacteria in the bird's intestine. Specifically, what are the genes IRS1, PRKAG2, PPP2R3C, PPM1K, GLUL, GADD45B, ZDHHC2. Have such correlations been reported before?

Re: Thank you for the comment. In order to fulfill our discussion part. Correlations between genes and microbiota were further investigated. And we added such discussions “In addition, as the results shown in figure 5, Lactobacillus sp. significantly correlated with IRS1. This correlation may further promote productive performance as nutritional ex-changes were a complicated process that required the assistant of energy and ion channel. The up-regulated genes of ATP7A, and ATP2A1, protein kinase, AMP activated gamma 2 (PRKAG2), and insulin receptor substrate 1(IRS1) after KLF supplement may provide more energy for nutrient exchanging process and therefore promoted the productive performance. ”  Please check Line 382-388.

I recommend the article for publication after revision in these parts.

Reviewer 3 Report

Comments and Suggestions for Authors

Manuscript ID: animals-3166910

Title: Gut-brain axial insight into the productive performances and egg quality responses to kudzu‑leaf flavonoids supplement of late-laying hens

The title is great. The concept of this article is fine and authors did many works with this research. The manuscript needs some revisions, because there are some aspects of the work that should be corrected and improved. Please, review the following recommendations:

- Introduction is very vague. It should be modified and be focused.

- All tables presented by the authors should be “self-explanatory” for simple understanding with well-clearly defined abbreviations as footnotes

- Discussion seems to be poor, didn't give good explanations of the results obtained. I think that it must be really improved. The discussion needs to be improved by adding bibliography to confirm the results obtained. Argue the discussion well, comparing the results obtained with articles reporting similar work by showing the performance obtained in the other papers.

- At the end of discussion, add a paragraph on describing the limitations of this work.

- Strengthen" 5. Conclusions" section  

- Line 15: Change "Results indicated" to "Results indicated that"

- Line 15: Write the full name before the abbreviation "KLF" to clarify it. And what is the difference between it and the previous abbreviation "KL" in line 14?

- Lines 19 & 20: Delete "(FE)" because you will not use it in this section.

- Lines 24 & 105 " A total of …………….. were  " were or was is correct?

- Lines 25 & 106: Add space after each "," in "0.2%,0.4%,0.6%,0.8%,1.0%"

- Line 27: Change "egg quality egg weight," to "egg quality, egg weight,"

- Lines 24-28: Authors did not specify the number of replicates used in each treatment.

- Lines 30-32: Rewrite this sentence

- In all text: Please delete all the pronouns like we, our, etc. throughout the manuscript and change the text appropriately. 

- Line 37: Add "that" after "indicated"

- Line 104: " …. and Birds Feeding procedure"?

- Line 113: Change "Each bird" to "Each hen"

- In Table 1: In the first row, change "composition" to "Composition%"

- In Table 1: Check the percentage of phosphorus in the experimental feed.

- In Table 1: Add the percentage of crude fat and crude fiber to the experimental feed.

- Line 119: Do you mean the "vitamin content" or the premix content (including vitamins and minerals)?

- Line 122: Change "Egg Quality Measurement" to "Egg Quality Measurements"

- Lines 125-126: The author mentioned that he estimated the amount of food to estimate some measures such as "egg production rate, deformity rate, average egg weight ", what is the relationship between them?. This sentence needs to be edited.

- In Table 2:  Add the period in days in each phase in the first column.

- In Table 2:  The values ​​of ADFI are different from what you mentioned in line 114 "daily feed allowance of 110 g", why?. Additionally why do the values ​​change between transactions even though you fixed the quantity?

- Line 207: Change "result are" to "results are"

- Line 208: Delete Space before "Whereas"

- In Table 3: P-value in "Haugh Unit" is 0.095 (> 0.05) & P-value in "Eggshell strength(N)" is 0.034 (< 0.05). The authors mentioned that there are significant differences in Haugh Unit parameter and wrote letters (a,b) to clarify the  significant differences. However, the author ignored the significant differences between the treatments in Eggshell strength(N) parameter. How is that?

- Lines 344-345: Rewrite this sentence

- Line 353: Delete "(GB)" because you will not use it again.

- Line 361: Delete "(CNS)" because you will not use it again.

- Insert the correct format style for journals in the references in the text and references list.

Comments on the Quality of English Language

Minor editing of English language required.

Author Response

Title: Gut-brain axial insight into the productive performances and egg quality responses to kudzu‑leaf flavonoids supplement of late-laying hens

The title is great. The concept of this article is fine and authors did many works with this research. The manuscript needs some revisions, because there are some aspects of the work that should be corrected and improved. Please, review the following recommendations:

- Introduction is very vague. It should be modified and be focused.

Re: Thank you for the comment. Based on the comment, we corrected the introduction to make it more logistical. The aim of the study was to investigate the underlying mechanism of the improving effects after KLF supplement through investigating the gut–brain axis and the underlying pathways. We hope these changes meet the requirement.  

- All tables presented by the authors should be “self-explanatory” for simple understanding with well-clearly defined abbreviations as footnotes

Re: Thank you for the comment. The footnote of each table was completed and we hope these changes meet the requirement.

- Discussion seems to be poor, didn't give good explanations of the results obtained. I think that it must be really improved. The discussion needs to be improved by adding bibliography to confirm the results obtained. Argue the discussion well, comparing the results obtained with articles reporting similar work by showing the performance obtained in the other papers.

Re: Thank you for the comment. In order to fulfill our discussion part. Correlations between genes and microbiota were further investigated. And we added such discussions “In addition, as the results shown in figure 5, Lactobacillus sp. significantly correlated with IRS1. This correlation may further promote productive performance as nutritional ex-changes were a complicated process that required the assistant of energy and ion channel. The up-regulated genes of ATP7A, and ATP2A1, protein kinase, AMP activated gamma 2 (PRKAG2), and insulin receptor substrate 1(IRS1) after KLF supplement may provide more energy for nutrient exchanging process and therefore promoted the productive performance. ”  Please check Line 382-388.

- At the end of discussion, add a paragraph on describing the limitations of this work.

Re: Thank you for the comment. The description has been added “Although the promotive effects significantly displayed after KLF supplement, further investigations were also required to fulfill the modulatory mechanism of KLF on late-laying hens. Functional analysis of the significantly proliferated bacterial communities should be studied through the culturation method, and functions of the differential expressed genes through cell-cultural method. Therefore, modulatory effects of KLF could be verified.” Please check line 403-408.

- Strengthen" 5. Conclusions" section  

Re: Thank you for the comment. Conclusion has been corrected as “Findings of the present study indicated KLF supplement effectively improved the productive performance and egg quality through significantly proliferated probiotics such as Bifidobacterium sp., and Lactobacillus sp., which may further interact with hypothalamus genes, and therefore promoted body health and productive performance.” Line 410-413.

- Line 15: Change "Results indicated" to "Results indicated that"

Re: Thank you for the comment. It has been corrected. Line 15.

- Line 15: Write the full name before the abbreviation "KLF" to clarify it. And what is the difference between it and the previous abbreviation "KL" in line 14?

Re: Thank you for the comment. KLF was short for “ Kudzu-leaf flavonoid”. It has been added in Line 15.

- Lines 19 & 20: Delete "(FE)" because you will not use it in this section.

Re: Thank you for the comment. (FE) has been deleted. Please check Line 20 and line 21.

- Lines 24 & 105 " A total of …………….. were  " were or was is correct?

Re: Thank you for the comment. 360 layer-hens were divided, and I think it is not a entirety. So we use “were” here. 

- Lines 25 & 106: Add space after each "," in "0.2%,0.4%,0.6%,0.8%,1.0%"

Re: Thank you for the comment. Spaces have been added. Please check Line 26.

- Line 27: Change "egg quality egg weight," to "egg quality, egg weight,"

Re: Thank you for the comment. It has been corrected. Line 29.

- Lines 24-28: Authors did not specify the number of replicates used in each treatment.

Re: Thank you for the comment. Sentence of “Each treatment contained 6 replicates, with 10 hens in each replicate” has been added in Line 27.

- Lines 30-32: Rewrite this sentence

Re: Thank you for the comment. Sentence has been corrected as “KLF supplement significantly decreased deformity rate while significantly increased the egg shell strength in the finishing phase after (P<0.05). Specifically, supplemented with 0.6% of KLF possessed the lowest deformity egg rate”. Line 32-34.

- In all text: Please delete all the pronouns like we, our, etc. throughout the manuscript and change the text appropriately. 

Re: Thank you for the comment. We thoroughly corrected the pronouns and hope these changes meet the requirement.

- Line 37: Add "that" after "indicated"

Re: Thank you for the comment. Change has been done. Line 39.

- Line 104: " …. and Birds Feeding procedure"?

Re: Thank you for the comment. These words have been removed. Line 104.

- Line 113: Change "Each bird" to "Each hen"

Re: Thank you for the comment. Change has been done. Line 113.

- In Table 1: In the first row, change "composition" to "Composition%"

Re: Thank you for the comment. Change has been done. Please check Table 1.

- In Table 1: Check the percentage of phosphorus in the experimental feed.

Re: Thank you for the kindly reminding. The phosphorus percent was 1% while the CaCO3 was 9.00%.

- In Table 1: Add the percentage of crude fat and crude fiber to the experimental feed.

Re: Thank you for the kindly reminding. The percentage of crude fat and crude fiber have been added. Please check Table 1.

- Line 119: Do you mean the "vitamin content" or the premix content (including vitamins and minerals)?

Re: Thank you for the kindly reminding. We tend to give the premix content. Please check Line 119.

- Line 122: Change "Egg Quality Measurement" to "Egg Quality Measurements"

Re: Thank you for the comment. Change has been done. Please check Line 122.

- Lines 125-126: The author mentioned that he estimated the amount of food to estimate some measures such as "egg production rate, deformity rate, average egg weight ", what is the relationship between them? This sentence needs to be edited.

Re: Thank you for the comment. All these parameters are indicators of feed efficiency. Therefore, the sentence has been corrected as “ In addition, the remaining feed mass was measured to determine egg production rate (EPR), deformity rate (DR), average egg weight (AEW), average daily feed intake (ADFI), and feed-to-egg ratio (FER) to determine the benefits of KLF supplement on feed efficiency.” Please check Line 125-128.

- In Table 2:  Add the period in days in each phase in the first column.

Re: Thank you for the comment. The days has been added. Please check Table 2.

- In Table 2:  The values ​​of ADFI are different from what you mentioned in line 114 "daily feed allowance of 110 g", why?. Additionally why do the values ​​change between transactions even though you fixed the quantity?

Re: Thank you for the comment. The feed intake was acquired by the discrepancy between the daily feed provision and the residue. In the actual experimental time, the residue was always differed. So that the ADFI is a little difference between each other.

- Line 207: Change "result are" to "results are"

Re: Thank you for the comment. Change has been done. Please check Line 213.

- Line 208: Delete Space before "Whereas"

Re: Thank you for the comment. Change has been done. Please check Line 214.

- In Table 3: P-value in "Haugh Unit" is 0.095 (> 0.05) & P-value in "Eggshell strength(N)" is 0.034 (< 0.05). The authors mentioned that there are significant differences in Haugh Unit parameter and wrote letters (a,b) to clarify the  significant differences. However, the author ignored the significant differences between the treatments in Eggshell strength(N) parameter. How is that?

Re: Thank you for the comment, and sorry for giving a wrong description. Haugh unit showed an increasing trend after all gradient KLF supplement(0.05<P<0.10). We deleted the letters behind each data of Haugh unit. Please check Table3.

- Lines 344-345: Rewrite this sentence

Re: Thank you for the comment. Sentence has been corrected as “Moreover, flavonoids supplementation may also improve the egg quality through promoting the body health and benefiting the intestinal health of layer hens”. Line 352-353

- Line 353: Delete "(GB)" because you will not use it again.

Re: Thank you for the comment. Change has been done. Please check Line361.

- Line 361: Delete "(CNS)" because you will not use it again.

Re: Thank you for the comment. Change has been done. Please check Line369.

- Insert the correct format style for journals in the references in the text and references list.

Re: Thank you for the comment. All references have been corrected according to MDPI standards, Please check the references part.

Reviewer 4 Report

Comments and Suggestions for Authors

Gut-brain axial insight into the productive performances and egg quality responses to kudzu-leaf flavonoids supplement at late-laying hens

Dear Authors,

The manuscript is interesting, and well prepared. Describing effect of flavonoids from kudzu-leaf on performance, quality of eggs and microbiome of late-laying hens.

Some corrections are required before further processing of the manuscript.

Below I add some suggestions helpful in this process:

Line 2

Text in the title of manuscript must be justify.

Line 16

Genus name, sp. or spp. must be added: Bifidobacterium sp. or spp.

Lines 33-34 and 38

Sp. or spp. required in case of genus name.

Lines 46-382

Space before reference is required. ‘…layer hens [1].

Line 87

Subsection title can starts from capital letter ‘Kudzu’.

Line 95

Please add number of samples in each treatment.

Line 102

Please centre the equation.

Line 118

Table 1

Maybe in this place better will be title: Ingredient and chemical composition of experimental diets. Composition can be merged with Ingredient, as Ingredient composition, and in second column amount or other description can be added.

In second part of table Chemical composition row can be added before metabolizable energy content.

Abbreviations must be expanded in the table, or explained/described under the table (SBM, ME, CP,dM+C).

Line 119

Space before unit is required.

Line 144

Year of publication (2011) in case of this reference is not required, please delete, number of reference is sufficient.

Line 177

Student-Newman-Keuls is post-hoc test conducted after one-way ANOVA test, that can be also emphasized in case of statistical analysis.

Line 202

Space before unit required.

Line 203

Information about significance level must be added under the table. Letters a, b in each row describes significant differences between treatments at p<0.05.

Lines 189-316

p-value instead of P-value must be used (samples).

Line 214

Eggshell strength (N) p-value point at significant differences between treatments, post-hoc test must be conducted here or by mistake significance level description is added in next row Haugh unit (p = 0.095).

Information about significance level must be added under the table. Letters a, b in each row describes significant differences between treatments at p<0.05.

Line 265

Post-hoc test for Chao1 (b, a). This description of significance level is possible to present (n = 10 in treatment).

Information about significance level must be added under the table. Letters a, b in each row describes significant differences between treatments at p<0.05.

Line 287

Description of significance level for different genera, where p < 0.05.

Information about significance level must be added under the table. Letters a, b in each row describes significant differences between treatments at p<0.05.

Lines 330 and 352

Number of subsection respectively 4.1 and 4.2 must be added.

Line 386

‘…supplement significantly (p < 0.05)…’ can be added.

Line 387

.sp must be added

Lines 409-477

References

Must be adapted to Animals/MDPI standards described in Instructions for Authors (abbreviation of Journal’s name with points/dots, italics in case of Journal’s name and volume/issue, doi link)

I.e. reference no.1

1. Anene, D.O.; Akter, Y.; Thomson, P.C.; Groves, P.; Lin, S.; O’Shea, C.J. Hens that exhibit poorer feed efficiency produce eggs with lower albumin quality and are prone to being overweight. Animals 2021, 11(10), 2986. https://doi.org/10.3390/ani11102986

Author Response

Dear Authors,

The manuscript is interesting, and well prepared. Describing effect of flavonoids from kudzu-leaf on performance, quality of eggs and microbiome of late-laying hens. Some corrections are required before further processing of the manuscript. Below I add some suggestions helpful in this process:

Line 2

Text in the title of manuscript must be justify.

Re: Thank you for the comment, and the change has been done. Line 2.

Line 16

Genus name, sp. or spp. must be added: Bifidobacterium sp. or spp.

Re: Thank you for the comment. Bifidobacterium sp., and Lactobacillus sp., have been corrected. Please check Line 16.

Lines 33-34 and 38

Sp. or spp. required in case of genus name.

Re: Thank you for the comment. Sp. Added in every genus name. Please check Line 35-38.

Lines 46-382

Space before reference is required. ‘…layer hens [1].

Re: Thank you for the comment. Spaces before references have been added.

Line 87

Subsection title can starts from capital letter ‘Kudzu’.

Re: Thank you for the comment. Kudzu has been capital. Please check Line 88.

Line 95

Please add number of samples in each treatment.

Re: Thank you for the comment. 10mg of flavonoids standard samples (n=6) were dissolved into 50ml ethanol. Line 96.

Line 102

Please centre the equation.

Re: Thank you for the comment. Change has been done. Please check Line103.

Line 118

Table 1

Maybe in this place better will be title: Ingredient and chemical composition of experimental diets. Composition can be merged with Ingredient, as Ingredient composition, and in second column amount or other description can be added.

In second part of table Chemical composition row can be added before metabolizable energy content.

Abbreviations must be expanded in the table, or explained/described under the table (SBM, ME, CP,dM+C).

Re: Thank you for the comment. The title has been corrected as “Ingredient and chemical composition of experimental diets.”. We added the composition amount (%) in the second column, and the explain of Abbreviations was added in the table. Please check Table 1.

Line 119

Space before unit is required.

Re: Thank you for the comment. Change has been done. Please check Line120-122.

Line 144

Year of publication (2011) in case of this reference is not required, please delete, number of reference is sufficient.

Re: Thank you for the comment. Change has been done. Please check Line151.

Line 177

Student-Newman-Keuls is post-hoc test conducted after one-way ANOVA test, that can be also emphasized in case of statistical analysis.

Re: Thank you for the comment. The description has been corrected as “the comparison test was applied using one-way ANOVA, and the Stu-dent-Newman-Keuls model was following used for the post-hoc test to investigate the differences among treatments ” Please check Line 183-185.

Line 202

Space before unit required.

Re: Thank you for the comment. Change has been done. Please check Table 2.

Line 203

Information about significance level must be added under the table. Letters a, b in each row describes significant differences between treatments at p<0.05.

Re: Thank you for the comment. The description of significance, “ Letters a, b in each row describes significant differences between treatments at P<0.05.” has been added in Line 210.

Lines 189-316

p-value instead of P-value must be used (samples).

Re: Thank you for the comment. We corrected all “P-value” in each table.

Line 214

Eggshell strength (N) p-value point at significant differences between treatments, post-hoc test must be conducted here or by mistake significance level description is added in next row Haugh unit (p = 0.095).

Re: Thank you for the comment, and sorry for giving a wrong description. Haugh unit showed an increasing trend after all gradient KLF supplement(0.05<P<0.10). We deleted the letters behind each data of Haugh unit. Please check Table 3.

Information about significance level must be added under the table. Letters a, b in each row describes significant differences between treatments at p<0.05.

Re: Thank you for the comment. The description of significance, “ Letters a, b in each row describes significant differences between treatments at P<0.05.” has been added. Please check Table 2.

Line 265

Post-hoc test for Chao1 (b, a). This description of significance level is possible to present (n = 10 in treatment).

Information about significance level must be added under the table. Letters a, b in each row describes significant differences between treatments at p<0.05.

Re: Thank you for the comment. Significant symbols of Chao1 index have been added. The description of significance, “ Letters a, b in each row describes significant differences between treatments at P<0.05.” has been added. Please check Table 4.

Line 287

Description of significance level for different genera, where p < 0.05.

Information about significance level must be added under the table. Letters a, b in each row describes significant differences between treatments at p<0.05.

Re: Thank you for the comment. Significant symbols of Chao1 index has been added. The description of significance, “ Letters a, b in each row describes significant differences between treatments at P<0.05.” has been added. Please check Table 5.

Lines 330 and 352

Number of subsection respectively 4.1 and 4.2 must be added.

Re: Thank you for the comment. number of subsection 4.1 and 4.2 has been added.

Line 386

‘…supplement significantly (p < 0.05)…’ can be added.

Re: Thank you for the comment. Change has been done. Please check line 392.

Line 387

.sp must be added

Re: Thank you for the comment. Change has been done. Please check Line 393.

Lines 409-477

References

Must be adapted to Animals/MDPI standards described in Instructions for Authors (abbreviation of Journal’s name with points/dots, italics in case of Journal’s name and volume/issue, doi link)

I.e. reference no.1

  1. Anene, D.O.; Akter, Y.; Thomson, P.C.; Groves, P.; Lin, S.; O’Shea, C.J. Hens that exhibit poorer feed efficiency produce eggs with lower albumin quality and are prone to being overweight. Animals202111(10), 2986. https://doi.org/10.3390/ani11102986

Re: Thank you for the comment. All references have been corrected according to MDPI standards, Please check the references part.